

# Corona or hydrochloric acid modulates embryonic diapause in silkworms by activating different signaling pathways

Yuli Zhang[1], Guizheng Zhang[1], Mangui Jiang[1], Pingyang Wang[1], Xia Wang[1], Qiuying Cui[1] and Quan Sun[2]

[1] Guangxi Key Laboratory of Sericultural Genetic Improvement and Efficient Breeding, Sericultural Research Institute, Nanning, Guangxi Zhuang Autonomous Region, China
[2] Chongqing University of Posts and Telecommunications, Chongqing Key Laboratory of Big Data for Bio Intelligence, School of Life Health Information Science and Engineering, Nan'an, Chongqing, China

## ABSTRACT

**Background**. To adapt to environmental changes, diapausing silkworm eggs remain dormant during the early stages of embryonic development. Various methods have been used to terminate silkworm egg diapause and promote egg hatching.

**Methods**. To elucidate the molecular mechanisms by which corona and other treatments terminate silkworm egg diapause, we collected eggs at 1, 6, and 20 h after treatments and sequenced their transcriptomes.

**Results**. The results showed that both corona and hydrochloric acid (HCl) treatments effectively terminated diapause and promoted egg hatching, with corona treatment inducing faster hatching than HCl treatment. Differentially expressed gene (DEG) analysis revealed the presence of fewer DEGs at 1 h, with a marked increase observed at 6 and 20 h post treatment. Functional enrichment analysis showed that the FoxO signaling pathway was activated at 6 h, with more substantial gene expression changes observed after corona treatment. In addition, HCl treatment appeared to activate the heat shock protein and hormone-regulated pathways. Our study results provide a basis for further analysis of the molecular mechanisms underlying diapause termination in silkworm eggs.

Corresponding author
Quan Sun, sunquan@cqupt.edu.cn

## INTRODUCTION

Diapause, a state of developmental arrest, is observed in various animals and is characterized by low energy consumption. This state helps animals survive extreme environmental conditions such as cold, heat, and drought, increasing their chances of survival and reproduction (*Podrabsky & Hand, 2015*). Insects can undergo diapause at different developmental stages, including eggs, larvae, pupae, and adults (*Feng et al., 2012*; *Xu et al., 2004*).

Silkworm (*Bombyx mori*), an economically important insect domesticated in China for over 5,000 years, provides silk, a valuable textile raw material with increasing applications in various industries (*Xia et al., 2007*; *Xia et al., 2004*). Silkworms are a typical example of

egg-diapausing insects, which enter diapause during the late gastrula stage (*Nakagaki et al., 1991*). Environmental factors influence egg diapause in silkworm; in colder regions, eggs enter diapause and hatch only once or twice a year, whereas in warmer regions such as southern China, eggs generally bypass diapause, undergoing multiple reproductive cycles annually (*Liang et al., 2014*). After egg laying, silkworm embryos undergo rapid division and growth; however, in diapausing eggs, cell division slows down and the embryos remain in the G2 phase after 72 h, halting further development (*Nakagaki et al., 1991*). In silkworm production, diapausing eggs can resume development when exposed to certain chemical or physical stimuli, such as acid treatment or friction (*Gong et al., 2016*; *Zhang et al., 2022*).

Although various methods have been developed to prevent diapause, the underlying diapause molecular mechanisms remain poorly understood (*Sonobe et al., 1979*; *Yamamoto, Mase & Sawada, 2013*). Current evidence suggests that the diapause hormone, secreted by neurosecretory cells in the suboesophageal ganglion, helps regulate diapause in insects (*Yamashita, 1996*). In addition, the conversion of glucose to trehalose and glycerol has been shown to improve survival during diapause, with these substances being rapidly degraded in diapausing eggs. Genome-wide microarrays, RNA sequencing (RNA-seq), and liquid chromatography–mass spectrometry have also been used to compare diapause and non-diapause eggs (*Akitomo et al., 2017*; *Fan et al., 2013*; *Gong et al., 2016*; *Sasibhushan, Ponnuvel & Vijayaprakash, 2013*). However, it remains unclear whether various methods of diapause prevention share the same molecular mechanisms (*Chino, 1958*).

Here, we aimed to elucidate the molecular mechanisms through which corona treatment prevents diapause in silkworm embryos and compare the effects of corona treatment and conventional hydrochloric acid (HCl) treatment. Using high-throughput transcriptome sequencing, we analyzed gene expression at multiple time points during early embryonic development after treatment. The results of this study provide valuable insights into the identification of key genes and the molecular regulatory network involved in preventing diapause in silkworm eggs through corona treatment.

## MATERIALS AND METHODS

### Silkworm materials

In this study, the silkworm strain 7532 was bred by the Guangxi Zhuang Autonomous Region Sericulture Technology Promotion Station. Domesticated silkworms were reared until moths hatched. Male and female moths were allowed to self-cross for 4 h, after which female moths were placed on silkworm egg paper to lay eggs. The eggs were maintained at 25 °C for three days and then stored at 5 °C for further analysis.

### Silkworm egg treatment

After 50 days, silkworm eggs were removed from refrigeration and left at room temperature for 4 h before treatment. The eggs were divided into three groups (each group contains 120–150 eggs): (1) acid treatment at 47.2 °C for 5 min and 30 s (HCl relative density = 1.092) (*Zhao et al., 2012*), (2) corona treatment (12 KV, 1 min) (*Zhang et al., 2022*), and (3) control, which received no treatment. Samples were collected at 1, 6, and 20 h after

treatment (HAT) and immediately frozen in liquid nitrogen for further analysis. Two replicates were prepared for each treatment group.

## RNA extraction, library construction, and sequencing

Total RNA was extracted using a TRIzol reagent kit (Invitrogen, Carlsbad, CA, USA) according to the manufacturer's protocol. RNA quality was assessed using an Agilent 2100 Bioanalyzer (Agilent Technologies, Palo Alto, CA, USA) *via* RNAse-free agarose gel electrophoresis. mRNA was enriched using oligo(dT) beads, fragmented, and reverse-transcribed into cDNA using NEBNext Ultra RNA Library Prep Kit for Illumina (NEB # 7530; New England Biolabs, Ipswich, MA, USA). The cDNA fragments were purified, ligated with Illumina sequencing adapters, and size-selected using agarose gel electrophoresis. The ligation products were then subjected to polymerase chain reaction (PCR) amplification, and the resulting cDNA library was sequenced using the Illumina NovaSeq 6000 platform.

## RNA-seq data analysis

Adapter sequences and low-quality reads were removed from each dataset using fastp (version 0.18.0) (*Chen et al., 2018*). An index of the reference genome was constructed, and paired-end clean reads were mapped to this reference genome using HISAT2 (version 2.2.4) *via* the "-rna-strandness RF" option and other default parameters (*Kim et al., 2019*). Clean reads were then used for gene assembly and abundance calculations. The silkworm reference genome and annotation files were downloaded from the National Center for Biotechnology Information genome database (accession no. ASM15162v1) (*Xia et al., 2004*). The fragments per kilobase of transcript per million mapped reads (FPKM) values were calculated *via* StringTie (version 1.3.1) and HISAT2 (*Langmead & Salzberg, 2012*; *Pertea et al., 2016*). The statistical power of this experimental design, calculated in RnaSeqSampleSize was 0.84 (https://cqs-vumc.shinyapps.io/rnaseqsamplesizeweb/) (*Zhao et al., 2018*). Differentially expressed gene (DEG) analysis was conducted using DESeq (*Love, Huber & Anders, 2014*), and transcripts showing an absolute log2 (fold change) value of > 1 and a $p$-value of < 0.05 were considered differentially expressed. Time series expression patterns of the three groups were analyzed using Mfuzz (*Kumar & Futschik, 2007*). DEGs were further subjected to Gene Ontology functional analysis and Kyoto Encyclopedia of Genes and Genomes (KEGG) pathway analysis (*Boyle et al., 2004*; *Kanehisa et al., 2007*; *Kanehisa & Goto, 2000*). RNA-seq data are available at NCBI BioProject (accession number PRJNA1165327).

## Validation of gene expression patterns

Ten genes were selected for validation using quantitative reverse transcription-PCR (qRT-PCR). First-strand cDNA was synthesized using a TUREscript 1st Strand cDNA Synthesis Kit (Aidlab, Beijing, China). Gene-specific primers are listed in Table S1. Actin3 was used as an internal control. qRT-PCR was conducted using $2 \times$ SYBR$^{®}$ Green Master Mix (DBI, Langsing, MI, USA) and IQ5 (Bio-Rad, Hercules, CA, USA) with a cycling temperature of 60 °C and a single peak on the melting curve to obtain a single product according to the manufacturer's instructions. The 20 μL reaction volume consisted of forward and reversed primers (1 μL), $2 \times$ SYBR$^{®}$ Green Master Mix (10 μL), ddH2O (6 μL), and cDNA (2 μL).

Three technical replicates were prepared for each gene. Relative gene expression levels were calculated using the $2^{-\Delta\Delta Ct}$ method and visualized graphically. All the numerical data in figures are presented as mean $\pm$ standard deviation (SD) of three independent experiments. The obtained data were subjected to unpaired a two-tailed Student's $t$-tests using GraphPad Prism software (version 8).

## RESULTS

### Effects of corona and HCl treatments on diapause termination in silkworm eggs

The results showed that silkworm eggs began hatching on the ninth day (Fig. 1A). By the tenth day, over 80% of the corona-treated eggs had hatched, whereas approximately 5% of the HCl-treated eggs had hatched. Moreover, none of the untreated eggs had hatched. Although the hatching rate of HCl-treated eggs eventually reached that of corona-treated eggs, the incubation period for corona-treated eggs was relatively shorter (Fig. 1B). To explore the molecular mechanisms through which corona treatment terminates diapause, we conducted further transcriptomic analyses of early-stage corona-treated silkworm eggs.

### RNA-seq of corona-treated, HCl-treated, and control groups

RNA samples from the control and corona-treated groups were collected at 1, 6, and 20 HAT. For comparison, RNA samples from the HCl-treated group were collected at 6 and 20 HAT. These samples were defined in Table 1 and sequenced using Illumina sequencing (Table 1). In total, approximately 752 million raw reads were obtained, with an average of 47 million reads per sample (Table S2). After filtering, >94% of the reads were classified as clean (Table S3). The clean reads were mapped to the silkworm reference genome, with a mapping rate of > 84% (Table S4). Over 64% of the mapped reads corresponded to genes, with > 95% mapped to exons and approximately 30% mapped to intergenic regions (Table S5).

### DEGs in corona-treated, HCl-treated, and control groups

First, we analyzed the correlation among corona-treated, HCl-treated, and control samples. The results showed a high degree of correlation between corona-treated samples at 1 h and control samples (at 1, 6, and 20 h), indicating that there was almost no difference between the corona-treated eggs of 1 HAT and control samples. E6 showed higher similarity to H20 and E20 than to H6 and CK20. The high similarity between E20 and H20 indicated that both the corona and HCl treatments after 20 h can activate the expression of genes required for embryonic development, potentially initiating the development process (Fig. 2A).

Comparisons between the treated and control groups showed relatively low numbers of DEGs in CK1 *vs.* E1 (109 DEGs, with 70 upregulated and 39 downregulated genes) and H20 *vs.* E20 (62 DEGs, with 34 upregulated and 28 downregulated genes) (Tables S6 and S7). However, at 6 HAT, 1,168 DEGs (502 upregulated and 666 downregulated genes) were observed in CK6 *vs.* E6 and 809 DEGs (503 upregulated and 306 downregulated genes) in CK6 *vs.* H6 (Tables S8 and S9). At 20 HAT, 1,116 DEGs (468 upregulated and 648 downregulated genes) were detected in CK20 *vs.* E20 and 1,351 DEGs (531 upregulated
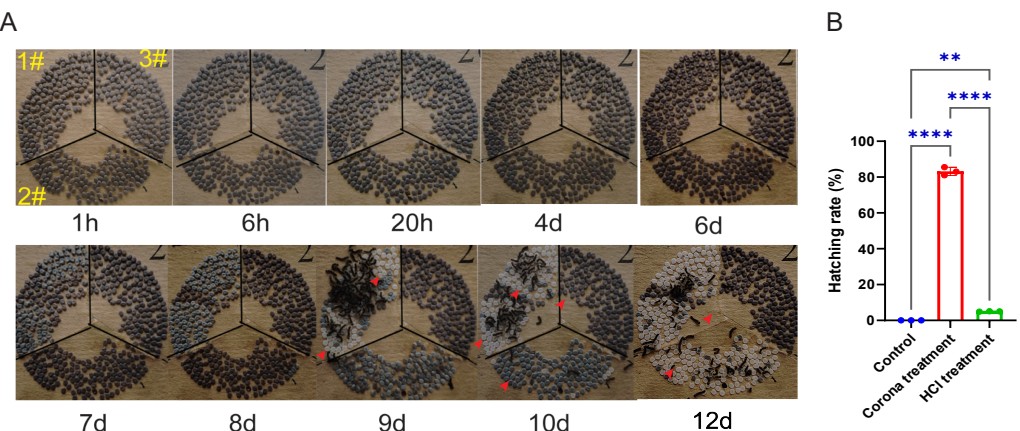

**Figure 1 Comparison of the effects of corona and HCl treatments on silkworm eggs.** (A) Images of eggs at different developmental stages from various treatment groups. 1#, corona treatment; 2#, HCl treatment; 3#, control. Red arrows indicate eggshells left by larvae after hatching. (B) Hatching rates of larvae from different groups at 10 days. Error bars represent one standard deviation ($n = 3$; **$p < 0.01$; ****$p < 0.0001$).

**Table 1 List of samples.**

| Treatment | 1 h | 6 h | 20 h |
| --- | --- | --- | --- |
| No treatment | CK1 | CK6 | CK20 |
| Corona (Electrically) treated | E1 | E6 | E20 |
| HCl treatment | | H6 | H20 |

and 820 downregulated genes) in CK20 *vs.* H20 (Tables S10 and S11). The similarity between E6 and H20 or E20 was even higher than that between E6 and H6, indicating substantial differences between E6 and H6. In particular, 947 DEGs (293 upregulated and 654 downregulated genes) were observed in H6 *vs.* E6 (Fig. 2B).

## Functions of DEGs in multiple comparison groups

As samples treated after 1 h and control samples showed minimal differences and the control group showed limited variation across the three time points, our study focused on DEGs between the treated groups (at 6 and 20 h) and the control group. Functional analysis of DEGs in the four comparison groups showed that DEGs in CK6 *vs.* E6 were mainly enriched in FoxO signaling pathway and vitamin B6 metabolism (Fig. 3A). Conversely, DEGs in CK6 *vs.* H6 were mainly enriched in pathways related to longevity regulation, protein processing in the endoplasmic reticulum, and arginine and proline metabolism (Fig. 3B). At 20 HAT, DEG enrichment in CK20 *vs.* E20 and CK20 *vs.* H20 showed certain similarities. The main enriched pathways in both comparison groups included glycolysis/gluconeogenesis, vitamin B6 metabolism, and pentose phosphate pathway (Figs. 3C and 3D).
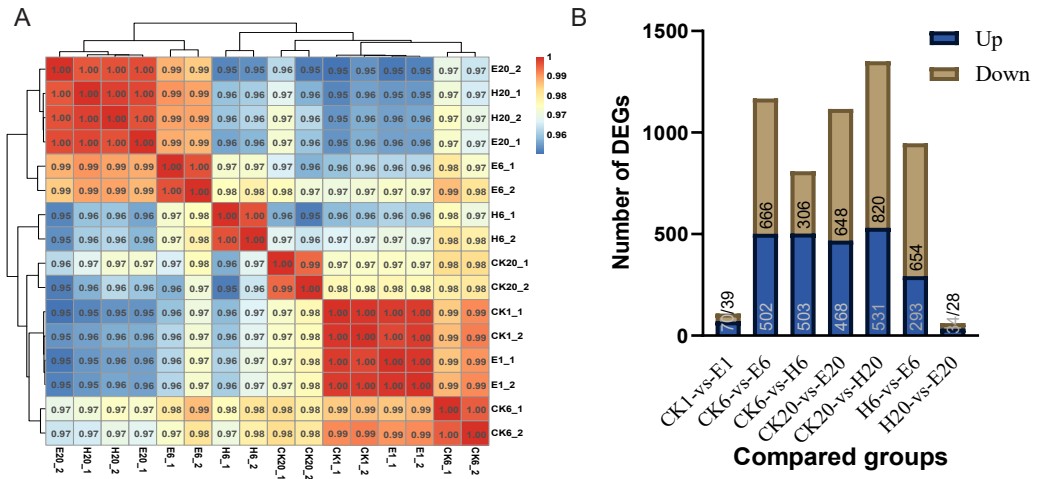

**Figure 2  Analysis of DEGs.** (A) Correlation coefficients between gene expression datasets. Red and blue colors indicate positive and negative correlation coefficients between samples, respectively. (B) Number of DEGs among the comparison groups.

## Identification of DEGs involved in corona and HCl treatments

Gene function enrichment analysis revealed differences in gene expression were observed between corona and hot HCl treatments at 6 h. A comparison of DEGs at multiple time points between the treated and control groups showed only nine common DEGs across five comparison groups and 129 common DEGs across four comparison groups (Figs. 4A and 4B). This suggests that corona and HCl treatments may modulate different pathways to promote silkworm embryonic development.

At 6 HAT, DEG analysis showed that two comparison groups—CK6 *vs.* E6 and CK6 *vs.* H6—shared 295 DEGs, and their functions were mainly related to cellular senescence. The CK6 *vs.* E6 comparison group had 873 specific DEGs, which were mainly enriched in purine metabolism, whereas the CK6 *vs.* H6 comparison group had 514 specific DEGs, which were mainly related to the longevity regulating pathway and protein processing in the endoplasmic reticulum (Fig. 4C).

At 20 HAT, the number of common DEGs between CK20 *vs.* E20 and CK20 *vs.* H20 increased to 888, with their functions enriched in the Toll and lmd signaling pathways and insect hormone biosynthesis. The CK20 *vs.* E20 comparison group had 228 specific DEGs, which were mainly associated with the p53 signaling pathway, whereas the CK20 *vs.* H20 comparison group had 463 specific DEGs, which were enriched in the purine metabolism pathway (Fig. 4D).

## FoxO signaling pathway response to corona treatment

After corona treatment, the FoxO signaling pathway was notably enriched. Corona treatment likely induces DNA damage, which may lead to the upregulation of *CDK2* and *Ras1*. *Ras1* promotes the expression of *FoxO* genes, whereas *CDK2* has been shown to inhibit the expression of downstream *FoxO* genes. The interplay between these factors may result in a complex regulatory effect on the overall expression of *FoxO* genes (Figs. 5A and

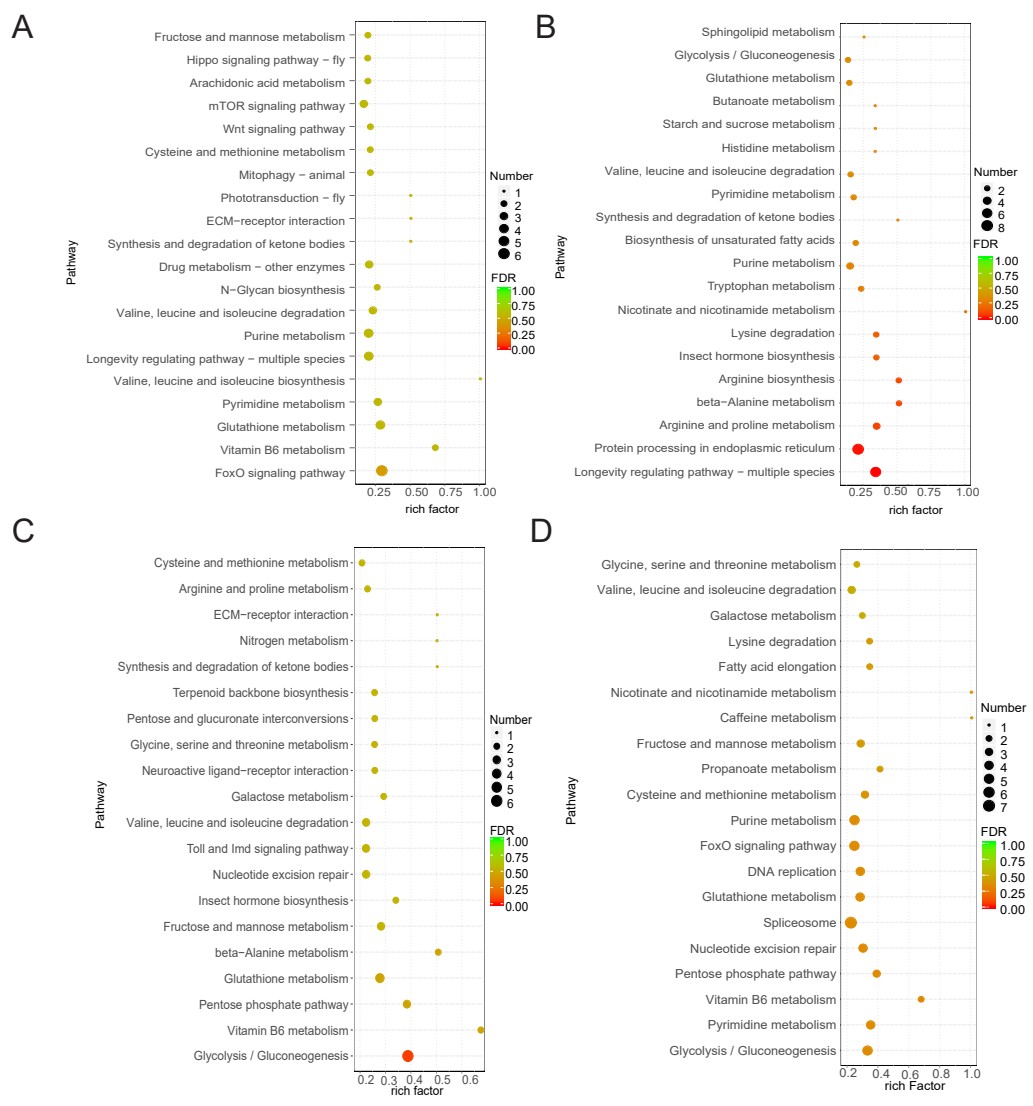

**Figure 3 Function annotation of DEGs.** (A) KEGG analysis of DEGs between CK6 and E6. (B) KEGG analysis of DEGs between CK6 and H6. (C) KEGG analysis of DEGs between CK20 and E20. (D) KEGG analysis of DEGs between CK20 and H20. The color and size of the bubbles indicate the level of significant enrichment and the number of genes, respectively.

5B). FoxO genes ultimately influence the expression levels of cell cycle–related genes, such as Cyclin B2/B3 (significantly upregulated after treatment) and Cyclin G2 (downregulated after treatment) genes (Figs. 5A and 5B). These results suggest that corona treatment activates cell cycle regulation through the FoxO signaling pathway. Further validation of corona-treated samples using qRT-PCR showed that the expression patterns of most genes, except for *Ras1*, were consistent with the RNA-seq data. In addition, several genes in the control group showed changes in expression levels across the three time points, which may be related to the process of adjusting the silkworm eggs to room temperature after removing them from refrigeration (Fig. 5C).

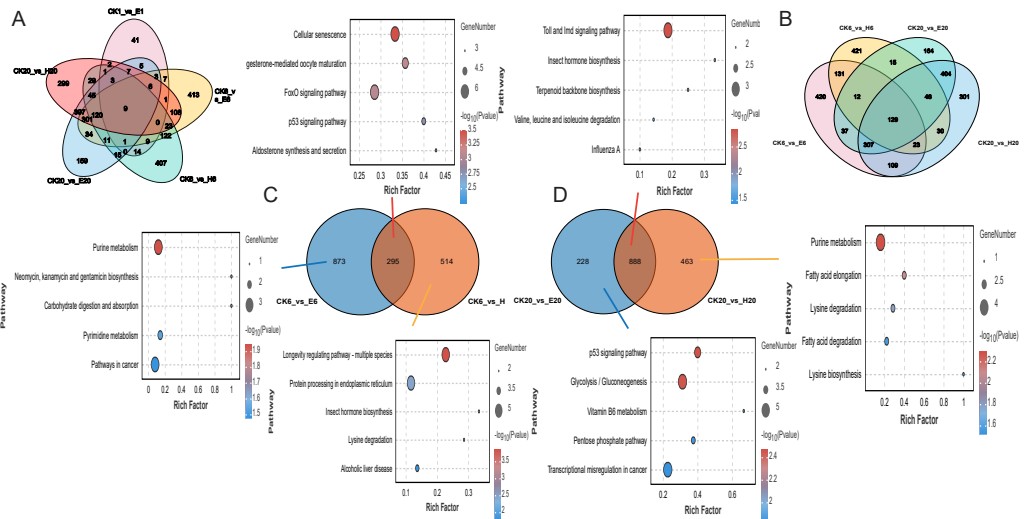

**Figure 4   Differentially expressed genes among different samples.** (A) Number of DEGs at multiple time points. (B) Number of DEGs at 6 and 20 h. (C) Number of DEGs at 6 h. (D) Number of DEGs at 20 h. KEGG analysis of DEGs.

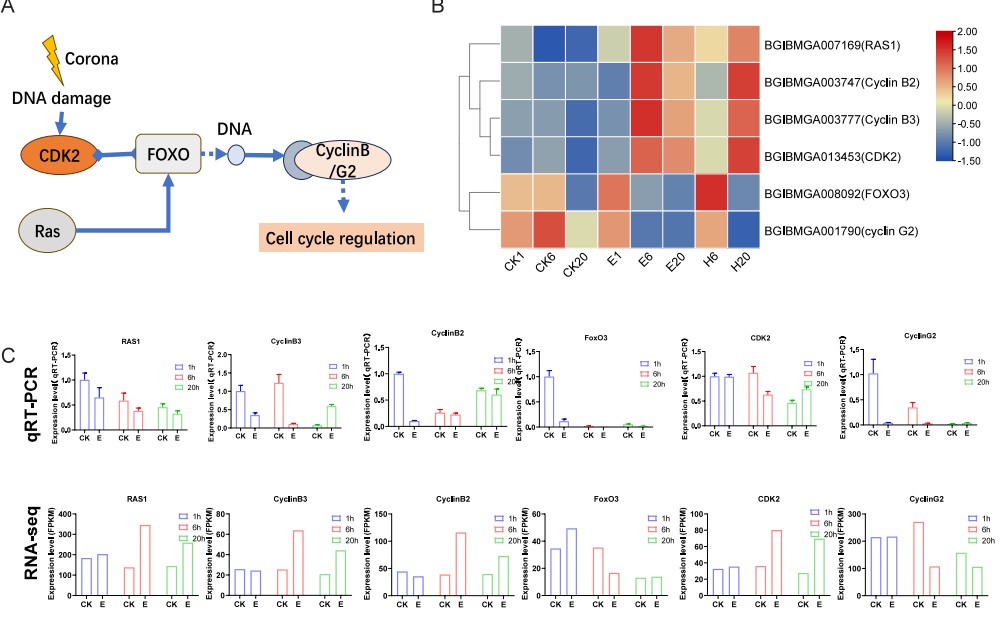

**Figure 5   FoxO signaling pathway.** (A) Schematic representation of the FoxO signaling pathway. (B) Expression profile of genes in the FoxO signaling pathway. Log2-scaled FPKM values are shown, ranging from low (blue) to high (red). (C) Gene expression profiles obtained *via* qRT–PCR were compared with RNA-seq data.

### Heat shock proteins and insect hormone biosynthesis may be involved in embryonic development *via* HCl treatment

The results showed that the enriched pathways in the HCl-treated group differed significantly from those in the corona-treated group. Notably, HCl treatment increased the expression of multiple heat shock protein (HSP)-related genes, potentially activating downstream pathways such as longevity regulation and ER-associated degradation (Figs. 6A and 6B). In addition, hormone-related pathways may be activated, with upregulated expression of *ALDH* and *CYP18A1* possibly promoting the production of juvenile and molting hormones (Figs. 6A and 6B). Examination of HCl-treated samples through qRT-PCR showed that the expression levels of *HSP70* and *ALHD* were significantly upregulated at 1 and 6 HAT. Conversely, the corona-treated group showed relatively minimal changes. Similarly, the expression levels of *sHSP20* and *CYP18A* showed significant upregulation at 6 HAT compared with those at 1 HAT. Fluctuations in gene expression levels in the control group were also observed, possibly due to temperature changes after removing the eggs from refrigeration (Fig. 6C). Although 1-h data from RNA-seq were unavailable, gene expression trends were generally consistent with qRT-PCR results.

## DISCUSSION

Silkworm, an economically important insect, undergoes diapause and other physiological adaptations to survive adverse conditions, such as cold and drought. Research has indicated a link between diapause tendencies in offspring and environmental conditions experienced by parental embryos (*Podrabsky & Hand, 2015*). In sericulture, various methods are used to artificially terminate diapause in silkworm eggs. These methods include traditional techniques such as acid immersion and refrigeration (*Zhao et al., 2012*) as well as treatments involving hydrogen peroxide (*Shen, Zhao & Liu, 2003*), dimethyl sulfoxide (*Yamamoto, Mase & Sawada, 2013*), and corona (*Zhang et al., 2022*). Despite these advancements, the molecular mechanisms underlying diapause termination in silkworms remain unclear, and it is unknown whether different termination methods share the same mechanism.

In this study, we showed that both corona and HCl treatments effectively terminated diapause in silkworm eggs. However, corona-treated eggs exhibited a shorter embryonic development period and earlier hatching than HCl-treated eggs. Notably, at 6 HAT, the corona-treated samples showed higher similarity to the 20 HAT samples of corona-treated and HCl-treated, suggesting that corona treatment induces earlier embryonic development. This further indicates that the mechanisms underlying diapause termination *via* corona and HCl treatments may differ.

When DEGs from corona and HCl treatments were analyzed for KEGG pathway enrichment, significant differences were noted. At 6 HAT, DEGs from the corona-treated group were enriched in the FoxO signaling pathway, whereas those from the HCl-treated group were mainly enriched in the longevity regulating pathway. This suggests a divergence in the early mechanisms of diapause termination. However, at 20 HAT, DEGs from both treatment groups showed significant enrichment in pathways such as glycolysis/gluconeogenesis, vitamin B6 metabolism, and pentose phosphate pathway,

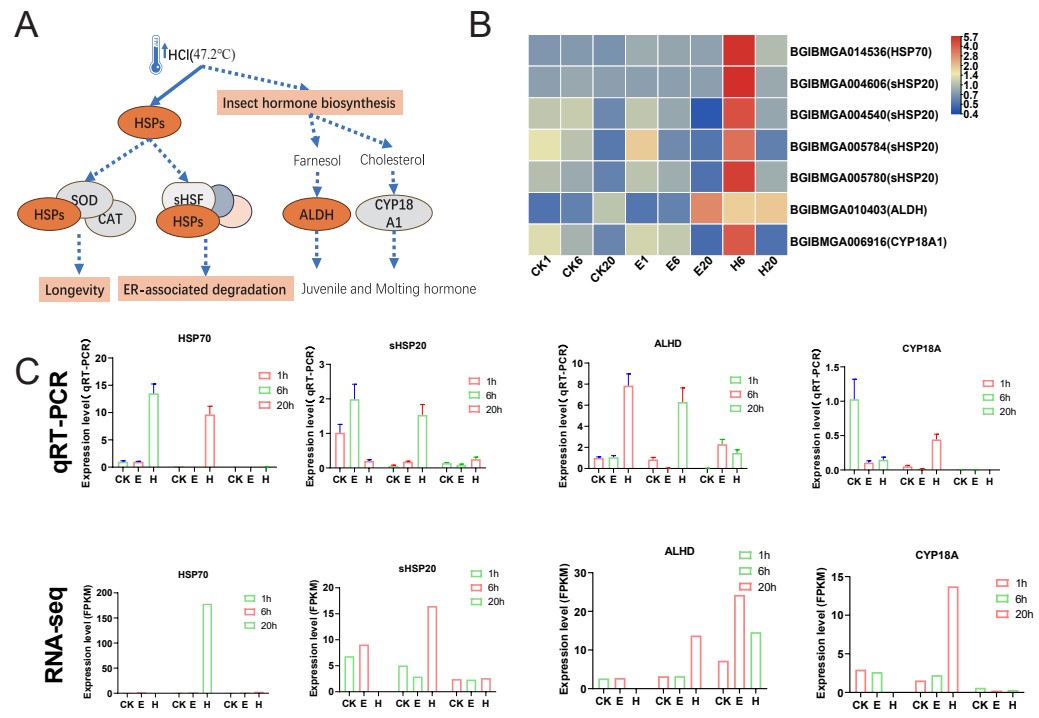

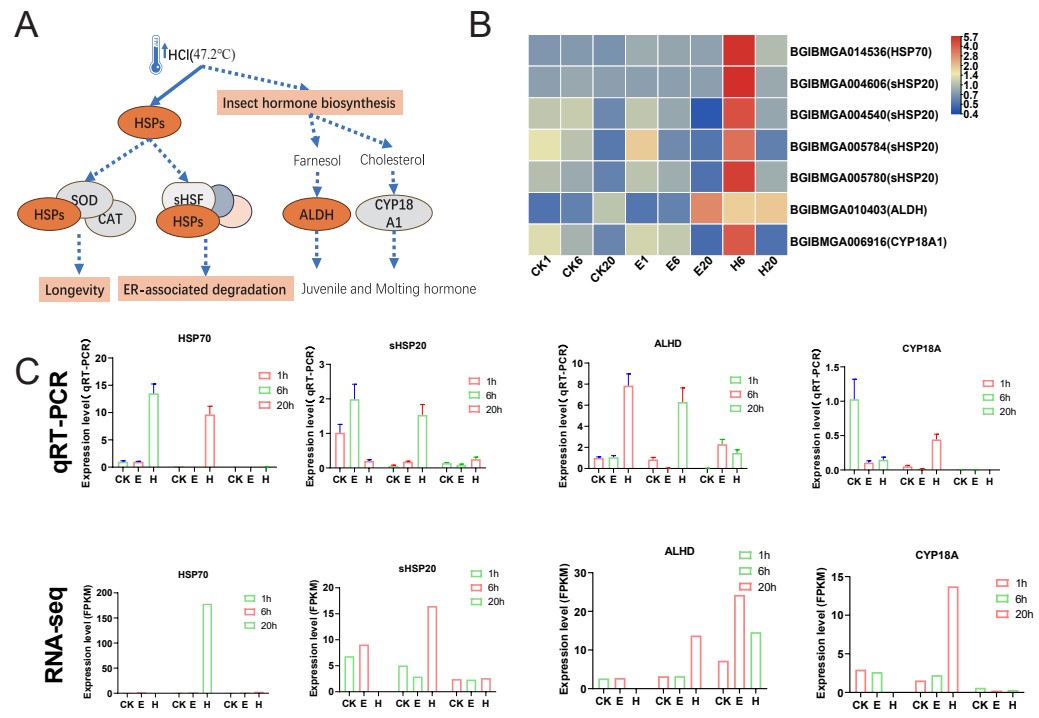

**Figure 6** **HSPs and insect hormone biosynthesis pathways.** (A) Schematic of HSPs and insect hormone biosynthesis pathways. (B) Gene expression patterns in HSPs and insect hormone biosynthesis. Log2-scaled FPKM values are shown, ranging from low (blue) to high (red). (C) Gene expression profiles obtained *via* qRT–PCR were compared with RNA-seq data.

all of which are fundamental to developmental processes and provide essential energy for embryonic development (*Lin & Xu, 2016*). Current evidence indicates that glycolysis is closely related to insect diapause, with the expression of HK—a key enzyme in the glycolytic pathway—being significantly higher in non-diapause *Drosophila* than in diapause *Drosophila* (*Castro-Sosa et al., 2017*). Our results indicated that after diapause termination, both corona- and HCl-treated silkworm eggs largely restore normal embryonic metabolism within 20 h, transitioning into the developmental stage. The 6-h post-treatment period represents a critical phase in the early regulation of gene expression during diapause termination, warranting further investigation.

As mentioned previously, the molecular mechanisms underlying diapause termination *via* corona and HCl treatments may differ, especially at 6 HAT. During corona treatment, the applied current damages DNA, leading to the activation of genes involved in the FoxO signaling pathway. FoxO plays a critical role in regulating the cell cycle, with its expression being regulated by genes such as *Ras* (*Kloet et al., 2015*). Conditional activation of FoxO factors has been linked to cellular proliferation and cell fate (*Lasick et al., 2023*; *Lei & Quelle, 2009*; *Schmidt et al., 2002*). In particular, during diapause regulation, FoxO may activate cell cycle-dependent kinase inhibitors to maintain the quiescent state of the cell cycle (*Karp & Greenwald, 2013*). In mosquitoes, at least 72 genes with FoxO binding sites have been identified, many of which are functionally associated with diapause

(*Sim & Denlinger, 2008*; *Sim & Denlinger, 2013*; *Sim et al., 2015*). The FoxO factors exert a negative regulatory effect on the cell cycle. Our study indicates that the downregulation of *FoxO3* in the treatment groups may play a role in initiating the development of silkworm embryos. While *RAS1* positively regulates *FoxO3* expression, the qRT-PCR results align with our expectations. However, the discrepancy between RNA-seq and qRT-PCR data suggests that the involvement of *RAS1* in *FoxO* expression regulation warrants additional experimental validation. We hypothesized that corona-induced DNA damage activates the FoxO regulatory pathway, thereby terminating silkworm embryo diapause.

Interestingly, hot HCl treatment may be associated with a different mechanism for diapause termination. In the present study, substantial activation of HSPs, including HSP70 and HSP20, suggesting that heat shock plays a dominant role in diapause termination process. The role of small HSPs in diapause has been investigated previously, with proteins such as ArHSP21 and ArHSP22 being expressed in diapause eggs and degraded upon diapause termination (*Qiu & MacRae, 2008a*; *Qiu & MacRae, 2008b*).

In this study, we also demonstrated that the method of cold storage and preservation of silkworm eggs may significantly affect gene expression. When the eggs were returned to room temperature, changes in gene expression were noted, raising the question of whether these genes are related to the survival of silkworm eggs in response to low temperatures. In addition, as RNA-seq data were not available for 1 h post HCl treatment, the 1-h post-treatment qRT-PCR data provided a valuable reference for further speculation on the timing of gene responses.

## CONCLUSION

This study compared corona and HCl treatments methods for terminating silkworm egg diapause using transcriptomic analysis. We revealed that the molecular mechanisms involved in diapause termination differ between the corona and HCl treatments. Corona treatment may regulate embryonic development through the FoxO signaling pathway, whereas HCl treatment may regulate it through the induction of HSPs and synthesis of *B. mori* hormones. These results provide a reference for further understanding the molecular mechanisms of diapause termination in silkworms and lay a theoretical foundation for breeding and production applications.

### Funding
This work was supported by National Natural Science Foundation of China (32360860). The funders had no role in study design, data collection and analysis, decision to publish, or preparation of the manuscript.

### Grant Disclosures
The following grant information was disclosed by the authors:
National Natural Science Foundation of China: 32360860.

## Competing Interests

The authors declare there are no competing interests.

## Author Contributions

- Yuli Zhang conceived and designed the experiments, performed the experiments, analyzed the data, prepared figures and/or tables, authored or reviewed drafts of the article, and approved the final draft.
- Guizheng Zhang performed the experiments, prepared figures and/or tables, and approved the final draft.
- Mangui Jiang performed the experiments, prepared figures and/or tables, and approved the final draft.
- Pingyang Wang performed the experiments, prepared figures and/or tables, and approved the final draft.
- Xia Wang performed the experiments, prepared figures and/or tables, and approved the final draft.
- Qiuying Cui performed the experiments, prepared figures and/or tables, and approved the final draft.
- Quan Sun analyzed the data, prepared figures and/or tables, authored or reviewed drafts of the article, and approved the final draft.

## Data Availability

The data are available at NCBI BioProject: PRJNA1165327.

## Supplemental Information

Supplemental information for this article can be found online at http://dx.doi.org/10.7717/peerj.18966#supplemental-information.

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
