# Peer review of "Corona or hydrochloric acid modulates embryonic diapause in silkworms by activating different signaling pathways"

_PeerJ, doi:10.7717/peerj.18966_

## Round 0.1 · original submission · Major Revisions

· Academic Editor

Major Revisions

The manuscript, as it stands, needs a major revision, and further clarification of the results. Indeed, there is no reference to where are the transcriptome data is deposited and available, which is a requirement for publication, and independent review. As stated by reviewer 1, the data needs to be analyzed further, as, for example, stating that ras1 is involved in the foxo pathway activation is not substantiated properly, as results are not consistent with that hypothesis, which happens to be one of the main tenets of the paper. The list of genes more differentially expressed is not furnished, making very difficult to validate the conclusions of the authors.

Reviewer 1 ·

Basic reporting

This manuscript demonstrated possible mechanism underpinning diapause termination upon treatment with both corona and hydrochloric acid (HCl). Diapause eggs were first chilled with refrigeration for 50 days, kept at room temperature for 4 h, and then divided into three groups: continued at room temperature as controls, HCl treatment, as well as treatment with corona. They then collected eggs at 1, 6, and 20 h after each treatment, sequenced their transcriptomes, and examined various gene expression for validation. From the temporal analysis of various gene expression, they said that the present study provided valuable insights for diapause termination mechanism by corona treatment. However, no significant impact was provided for this reviewer after careful reading the current manuscript. They only determined the time-dependent effects on several gene expressions after treatment with either corona or HCl.

Experimental design

Not so good.

Validity of the findings

Not so good.

Additional comments

In addition, the title did not fit well with the whole experiment. In the experiment, they used both treatment with corona and HCl. However, in the title, they only said corona treatment without HCl. Why?
For the possible mechanism underling diapause termination upon treatment with corona and HCl, the authors said that the mechanisms between treatment with corona and HCl may be different. Corona treatment terminated diapause through the FOXO signaling pathways; HCl treatment promoted development through the induction of HSPs and hormone biosynthesis. Considering that HCl treatment terminated diapause only 1 or 2 days later than corona treatment, it may be impossible for the current hypothesis. In addition, the manuscript contains many other typos, awkward sentences and grammatical errors, and therefore needs significant editing. Fig. 1B only showed hatching rate, no incubation period for corona-treated eggs (L119).

Annotated reviews are not available for download in order to protect the identity of reviewers who chose to remain anonymous.

Reviewer 2 ·

Basic reporting

The authors compared the differential expressed genes (DEGs) of corona treated silkworm (Bombyx mori) eggs and HCL treated eggs by RNA-seq, and validated main DEGs by RT-qPCR. They found corona terminate diapause of silkworm eggs via FoxO signaling pathway, which differs from HCl treatment via HSPs and hormones synthesis pathways. This finding is helpful to reveal the molecular mechanism of diapause termination.

Experimental design

The experimental design is reasonable,.

Validity of the findings

The findings were well valided.

Additional comments

Revision suggestion.
1. Line 113-115: The first sentences are not need, they may be put behind line 119.
2. Line 115-116: “The results showed that silkworms from corona-treated eggs began hatching on the ninth day”, the sentence is wrong. It should be “The results showed that silkworm eggs from corona-treated began hatching on the ninth day”.
3. Line 134-135: “This finding indicates that there was almost no difference between the control samples at these time points, and the embryos did not exhibit any changes.” It is better to put this sentence right after the previous sentence, as a clause “indicating that there was almost no difference between the corona-treated eggs of 1 HAT and control samples, i.e the embryos did not exhibit any changes.”
4. Line 132-138: Here the authors did not give the meaning of E1, E6, E20; CK1,CK6, CK20 and H6, H20. As well as in fig 2.
5. Fig4 is not clear enough.
6. Line 162-163: This suggests that the two treatments prevent different egg diapause pathways. This sentence does not accurately convey the meaning of the experimental results.
7. Line 176-179: The analysis in this section is to somewhat contradictory and unconvincing.
8. Line 211: “In this study, we found that…” should be “we showed that…”.

Annotated reviews are not available for download in order to protect the identity of reviewers who chose to remain anonymous.

---

## Round 0.2 · Minor Revisions

· Academic Editor

Minor Revisions

Dear Sirs:

After reviewing your amended manuscript several typos are still present. This necessitates a detailed review process. I suggest using a grammar / spelling checker. Also, there are still issues with reviewer 2, that should be addressed. Lastly, I think that discussion needs to address the issue of the foxo versus ras pathways, as they both have been shown to affect growth in different scenarios.

Reviewer 2 ·

Basic reporting

The authors compared the differential expressed genes (DEGs) of corona treated silkworm (Bombyx mori) eggs and HCL treated eggs by RNA-seq, and validated main DEGs by RT-qPCR. They found corona terminate diapause of silkworm eggs via FoxO signaling pathway, which differs from HCl treatment via HSPs and hormones synthesis pathways. This finding is helpful to reveal the molecular mechanism of diapause termination.

Experimental design

The experimental design is reasonable.

Validity of the findings

The findings were well valided.

Additional comments

The authors said that tney have revised Fig. 4 to improve the quality. But in the revised manuscript there is no a picture contained. Please check.

---

## Round 0.3 · Minor Revisions

· Academic Editor

Minor Revisions

The manuscript is improved from previous versions, but it still needs amending: for example, on page 7 line 134 and page 8 line 146 "for" is inappropriate, it should be changed to "after"; e.g., treatments were for the specified times in the methods sections, not the hours implied in these parts of the text. Furthermore, there is still no link to the actual RNAseq raw data deposited in a pubic database, and that needs to be done.

---

## Round 0.4 · accepted · Accept

· Academic Editor

Accept

Dear Sirs:

The amended version of your manuscript is now acceptable for publication in PeerJ.